**COMMENT**

# Psychology needs more diversity at the level of Editor-in-Chief

Gerald J. Haeffel [1✉], Zhicheng Lin [2,3], Adeyemi Adetula [4,5], Ivan Vargas[6], Jenalee R. Doom[7], Yusuke Moriguchi[8], Ana J. Bridges[6] & Willie R. Cobb[1]

There are racial, gender, and geographical disparities for editors-in-chief in psychology. This is a problem, and many counter arguments are not persuasive. It is time for the field – and in the power of individuals - to implement suitable measures to make change happen.

The process for choosing a new editor-in-chief is akin to choosing a new pope (minus the white smoke). In our opinion, it is mysterious and non-transparent; it happens behind closed doors, and only the elite participate. And, according to the data, the outcomes of this process are inequitable.

The findings of reviews examining the composition of editorial boards are consistent—across psychology (and science more generally) editorial boards are mostly White and Western. In one of the largest reviews to date, Liu and colleagues[1] found "the proportion of female editors persisted at about half that of female scientists, and that the proportion of female editors-in-chief has consistently been even smaller." Importantly, the gender disparity for editors-in-chief could not be explained by meritocratic factors (e.g., career length, productivity, impact).

Disparities are also found for geography and race. An analysis of over one million papers handled by nearly 65,000 editors found that researchers from Asia, Africa, and South America had fewer editors than would be expected based on their share of authored publications[2]. Further, non-White researchers were underrepresented on editorial boards based on their share of authored publications with the greatest degree of underrepresentation for Black researchers.

In this article, we highlight why diversity at the level of editor-in-chief matters, recommend solutions, and address counterarguments. We focus on diversity in terms of race, gender, and geography because these factors have received the most empirical attention; however, diversity also includes sexual orientation, religion, language, gender identity, age, disability status, and socioeconomic status. Further, we focus on journals in which the editor-in-chief is an academic researcher rather than a full-time professional editor as these journals are focused on in the cited reviews. Finally, we acknowledge the efforts of those individuals working to mitigate problems in the peer-review structural system.

## Representation at the level of editor-in-chief matters
Diversity at the level of editor-in-chief is important for several reasons:

1. Editors-in-chief are the gatekeepers of published research. Editors-in-chief create policies that determine what constitutes "good" science. They influence how their editorial board will weigh novelty and importance, and as a result, what is published. And the ability to publish "affects who *wants* to stay, and who is *able* to stay, in research-focused careers"[3] (p. 7).

[1]University of Notre Dame, South Bend, USA. [2]University of Science and Technology of China, Hefei, China. [3]The Chinese University of Hong Kong, Shenzhen, China. [4]Université Grenoble Alpes, Grenoble, France. [5]Alex Ekwueme Federal University Ndufu-Alike, Ndufu-Alike, Nigeria. [6]University of Arkansas, Fayetteville, USA. [7]University of Denver, Denver, USA. [8]Kyoto University, Kyoto, Japan. ✉email: ghaeffel@nd.edu

2. Editors-in-chief are the face of science. Seeing diverse faces in positions of power—external signals of inclusion and acceptance—may help close the gaps in who pursues careers in science. As noted by Collins[4], "when deciding whether or not they belong in the STEM field, it is imperative that students see others that look like them to break any socialized belief barriers about possibilities of success" (p. 161). Reflective identity (i.e., seeing people who look like you) promotes belonging and engagement, which predict who pursues careers in science.

3. Editors-in-chief choose the associate editors and other board members. The editors-in-chief choose the associate editors (e.g., https://www.apa.org/pubs/journals/resources/publishing-tips/editorial-board; https://www.psychonomic.org/page/2023mceditorsearch) who also dictate the scope and progress of science because they also determine what gets published. Moreover, associate editors gain the editorial experience needed to become an editor-in-chief. As a result, an editor-in-chief's influence can last decades as their hand-picked associate editors often become the next scientific gatekeepers.

## Recommendations

We propose five solutions to increase diversity at the level of editor-in-chief:

1. Change the selection criteria. To maximize diversity in demographics and ideologies, we must stop choosing people from the same schools and same academic networks who espouse the same ideologies and same scientific values. This means defining excellence by criteria other than h-indexes, national prominence, titles (full-professor), and other historic (White, exclusive) measures of prestige. These are biased and invalid indicators of research *quality*, and there is no evidence that they are associated with better judgment or efficacy as editor-in-chief (see counterargument #3).

   To increase diversity, we recommend broadening the selection criteria[5]. In addition to considering scientific contributions like publications, criteria should also include the creation and sharing of software, data sets, measures, etc. Indirect scientific contributions should also be assessed such as the ability to elevate a scientific team, mentoring students, creating free statistical and methodological tools, and promoting transparent and replicable scientific findings. Although this approach may not result in more effective (i.e., better judgment, less biased) editors-in-chief on an individual level, it should increase and diversify the pool of "eligible" candidates and, in turn, reduce homogeneity in the kinds of biases held by the people in this position.

2. Do not allow the editor-in-chief to select the associate editors. Editors-in-chief tend to choose associate editors who they know, look like them, and are like-minded[6,7]. This selection bias decreases diversity and conserves the status quo. It creates a structural system in which the demographics, ideology, and methods of those in power are maintained over time. Homogeneity at the level of editors-in-chief creates homogeneity on the editorial boards, which creates homogeneity in what is published and, in turn, causes homogeneity in the theories, methods, and content of research. This homogeneity stifles scientific progress. Instead, we recommend using an independent committee to choose editors-in-chief and associate editors (using quality-based selection criteria).

We understand this will be an unpopular recommendation as editors-in-chief want control over their editorial team. But we believe science is stronger when homophily is reduced and adversarial collaborations are promoted. The peer review system is not working as intended (see counterargument #3), so it is worth pursuing new ideas even if unpopular.

3. Set limits on how many editorial boards a person can serve on at one time. Researchers can serve on multiple editorial boards simultaneously (i.e., be an associate editor at multiple journals at the same time). This means it is possible for a small homogenous group of researchers to have an excessively large influence across psychological science. We recommend that a person serve as associate editor for no more than one journal at a time and serve only once as an editor-in-chief. It is important to spread the wealth.

4. Compensate editorial board members for their time and contributions. We have proportionately fewer women and people of color at the level of faculty, and by diversifying our boards, we are asking for greater service contributions. This is problematic considering that racial and gender minorities in academia are disproportionately underpaid and may not have the privilege of doing this type of "in-kind" work. Academic publishing companies have large profit margins, suggesting that compensation is reasonable. In addition, universities should reduce service loads, provide course releases, and/or grant financial compensation for any faculty member serving as an editor-in-chief.

5. Monitor the system and report diversity metrics. Monitoring is needed to determine if the changes are working to create a more equitable field. Journals should report author and editorial board demographics, and the topics and sample populations of papers that are rejected, either with or without peer review annually (note that there may be legal constraints on the kind of information obtained depending on the country). Reporting should also include the percent of papers that include racially/ethnically/gender/geographically diverse samples. It may be useful to create a metric that evaluates journals on factors related to diversity (similar to the Transparency of Research Underpinning Social Intervention Tiers that measure journal adherence to the Transparency and Open Promotion guidelines; https://www.trustinitiative.org/).

## Anticipated counter arguments

1. The diversity problem is not due to bias, but a lack of eligible candidates. According to this argument, the diversity of editors-in-chief is proportional to the diversity of the pool of "eligible" candidates (i.e., full professors with high h-index and grants). It is not a matter of bias or prejudice, but a supply chain issue (i.e., a numbers game). This is a critical counter argument because it ostensibly rules out systemic bias as an explanation for the disparities and, therefore, reduces the need to mitigate potential biases in the selection process.

   There is at least some empirical data that disputes this counter argument. Two large systematic reviews[1,2] both indicate that the disparities in gender, race, and geography cannot be fully explained by differences in eligibility metrics. But for the sake of argument, let us assume there

is a supply chain issue. Does this rule out bias as an explanation? No.

It is a mistake to assume that bias does not exist simply because diversity at the top-level matches diversity at the level just below it. This is because there may be bias preventing groups of people from reaching all levels. Research shows that there is bias keeping non-White scholars and women from attaining metrics needed for editor-in-chief such as publication rate, h-index, grants, and networking opportunities. For example, Liu and colleagues[2] found that non-White scientists have significantly longer delays between manuscript submission and acceptance than White scientists, reducing their rate of publication and potential for citations. Further, once manuscripts are accepted, Black and Latinx scientists receive fewer citations than White scientists (accounting for textual similarity among papers). Liu and colleagues note that the disparity in citations for Black scientists has been *increasing* over the past decades. Similar effects have been found for women scientists[8]. Finally, research shows systemic racial disparities in obtaining external funding[9]. Taken together, there is strong evidence that science is not conducting a fair race.

2. The solution is to create a pipeline of diverse candidates. The pipeline approach is problematic for several reasons. First, it delays action; we must wait for the pipeline to fill. Second, there is little reason to believe that the pipeline will result in a significant increase in "eligible" candidates. As discussed earlier, the pipeline is leaky; there are widespread systemic barriers that make it more difficult for women and people of color to advance to the level of full professor[7]. And finally, the pipeline approach allows those currently in charge to choose and train the next generation of editors. It maintains the current power hierarchy and reinforces the dominant ideologies and ways of doing business. It creates a pipeline of sameness. A primary purpose of increasing diversity is to inject new ideas and structural policies into science. Training programs can stifle change and maintain the status quo.

3. Less "qualified" editors-in-chief will break peer-review. The validity of this argument relies on two assumptions. First is the idea that full-professors with high h-indexes, grants, awards, and experience are better at being editors-in-chief than those with less impressive metrics. There is no evidence to support this assumption. Use of these metrics is based on tradition, not empirical findings. They are associated with research quantity, but not necessarily research quality. Awards, titles, and high h-indexes do not bestow greater critical thinking skills or reduce biases. Human decision making is flawed regardless of fame and success[10].

The second assumption is that appointing less prominent or inexperienced researchers to editor-in-chief could break peer-review. Peer-review has been shown to be unreliable, unable to identify major errors in research or fraud, and it fails to prevent questionable research practices or the proliferation of non-replicable findings[11,12]. This calls into question the notion that the current system needs protection.

## Conclusion

We conclude with a controversial take. Anecdotally, several White male editors-in-chief have acknowledged the power and responsibility of their position, the role of structural racism in psychology, and their commitment to diversity. We do not doubt the sincerity of these comments. At the same time, there is irony in these statements. The same structural bias they oppose is at least partially responsible for the powerful position they hold. Moreover, no one is required to accept the position of editor-in-chief. It is a volunteer position. If one believes that there is bias in the selection process and diversity in representation is needed, then turn down the position and advocate for greater diversity at editor-in-chief —because equitable representation at the top matters.

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

## Author contributions

G.J.H., Z.L., A.A., I.V., J.R.D., Y.M., A.J.B. and W.R.C. all contributed to the writing and editing of this article.

## Competing Interests

The authors declare no competing interests.
