## [Peer Review File · Communications Psychology]

1st Aug 23

Dear Professor Haeffel,

Thank you for submitting your manuscript titled "Diversity at the Level of Editor-in-Chief Remains Elusive" to Communications Psychology.

I have carefully read your manuscript and am writing to let you know that it could potentially be suitable as a Comment in Communications Psychology. However, for the piece to be taken forward to peer-review and potential publication, you would need to undertake some substantive edits, which I will lay out below.

Your piece makes a crucial point, the lack of diversity of people in editor-in-chief roles, and - partly as a result - the lack of diversity across editorial boards in psychology. While the general, near-universal issue is well-established across many meta-scientific papers, the very valuable contribution your work makes is to highlight a) why this state of affairs is a problem, b) why common explanations may not serve as a valid excuse, and c) what actions societies/society journals could take to improve diversity. We would very much like to take the piece forward, but it would require revisions in two domains and I suggest extension in a third.

First, the work does not fit our Perspective format (as currently submitted). Instead, we would consider it as a Comment. A Comment is a much shorter call to action, usually about 1500 words and with a limit of 10 references. In the present case, we could make an exception and extend the word length to 2000 words and 15 references. Because Comments use a different referencing style from review and research articles, this change would be relatively straightforward to implement and we can assist you in the matter. Likewise, there are redundancies in the text that can easily be removed to create a shorter and tighter narrative. In a first instance, we would be happy to send to peer-review a much longer document, with a view on editing the piece into format once feedback points a path to publication.

Second, your manuscript makes a range of really important points (as highlighted above), but it uses recent developments at a specific society journal for topical framing. Focusing on a limited number of actors, rather than relying on the extensive research that exists and shows the issue to be universal (which is also cited), makes your criticism appear narrower than it is and opens you up to a biased misreading of your work. Your piece has the potential to be read as an important critique of - and forceful call to action regarding - the lack of diversity across editors in chief and editorial boards in our field, not limited to certain journals. As highlighted in the attached document, you would need to let go of these parts of the text.

Third, your criticism focuses on the issues and procedures at society journals/academic journals, but spares journals with professional editors or a combination of professional editors and academic board members from scrutiny. Although some procedures and aspects naturally differ, you might want to apply your critical lens and set of recommendations to this other type of journals as well.

I appreciate these changes are substantive. We shall hope to receive your revised version as soon as you are able to complete the suggested revisions.

If you anticipate a delay of more than four weeks, please let us know. Likewise, if you are not interested in submitting a suitably revised manuscript in the future please let me know so we can close your file. If you have any questions, please contact me.

Please use the link below when you are prepared to resubmit.

[link redacted]

Thank you for your interest in Communications Psychology.

Best regards,

Marike

Marike Schiffer, PhD

Chief Editor

Communications Psychology

9th Oct 23

Dear Professor Haeffel,

Thank you for your patience during the editorial evaluation and peer-review process.

Your manuscript titled "Diversity at the Level of Editor-in-Chief Remains Elusive in Psychological Science" has now been seen by 2 reviewers, and I include their comments at the end of this message.

Based on the reports and our own reading of your manuscript, we are very interested in the possibility of publishing your Comment in *Communications Psychology*, but would like to consider your response to a list of concerns in the form of a revised manuscript before we make a decision on publication.

Comments are opinion pieces and naturally, the Reviewers agree with some aspects of your piece, and disagree with others. For, the key to a successful revision of an opinion piece is that authors use the critique constructively. We do not ask authors to change the opinion they express. Instead, responding constructively to referee reports means strengthening the piece where the referees pointed to weaknesses in the argument; focusing on arguments that are fair and compelling; expanding on aspects that referees confirmed to be interesting; and toning down or removing statements that are detrimental to the persuasiveness of the text.

Reviewer #2 highlights in particular one aspect, which incidentally aligns with the journal's general guidance: opinion pieces, including those that are controversial and/or concerned with problems in our field, are most useful and best received when they focus on solutions. While it is important to name grievances, persuasive, perhaps bold, proposals are what attract readers' attention and inspire lasting change. Both reviewers offer some feedback on your proposal. We ask you to shift your focus even more to a proposal for change and as you do so, to incorporate this feedback.

Reviewer #1 highlights that some statements are not persuasive without additional evidence and Reviewer #2 likewise finds that some allegories are not compelling. Where there is clear evidence for a statement of fact, and that evidence is not widely known, it makes sense to provide a reference. The present piece is Comment, not a review article (there is a limit of 10 references for your piece), it is therefore also permissible to highlight the opinion character of specific statements, rather than offering a citation. However, the opinion needs to be flagged as such and be worded precisely, making potential caveats explicit. Reviewer #1 also highlights that some of the issues you describe are not unique to psychology. This is an insight worth keeping in mind, however, please note that there is no requirement to limit yourself to examples taken from psychology.

In sum, we invite you to revise your manuscript taking into account all reviewer and editor comments.

EDITORIAL POLICIES AND FORMATTING

You will find a complete list of formatting requirements following this link:

<https://www.nature.com/documents/commsj-style-formatting-checklist-comment.pdf>

Please use the checklist to prepare your manuscript for resubmission.

If you have any questions about any of our policies or formatting, please don't hesitate to contact me.

Please use the following link to submit your revised manuscript and a point-by-point response to the referees' comments (which should be in a separate document to any cover letter):

[link redacted]

We hope to receive your revised paper within 4 weeks; please let us know if you aren't able to submit it within this time so that we can discuss how best to proceed. If we don't hear from you, and the revision process takes significantly longer, we may close your file.

We understand that due to the current global situation, the time required for revision may be longer than usual. We would appreciate it if you could keep us informed about an estimated timescale for resubmission, to facilitate our planning. Of course, if you are unable to estimate, we are happy to accommodate necessary extensions nevertheless.

Please do not hesitate to contact me if you have any questions or would like to discuss these revisions further. We look forward to seeing the revised manuscript and thank you for the opportunity to review your work.

Best wishes,

Marike

Marike Schiffer, PhD
Chief Editor
Communications Psychology

REVIEWERS' COMMENTS:

Reviewer #1 (Remarks to the Author):

I have read your comment with great interest. The title is directed towards psychology but the whole

content of the comment can be discussed in any science. I agree with most of your points but I think some of them have to be supported with more evidence. I suggest major revision, you may find my comments below:

1. The first sentence is alluding that the process of choosing editor in chief in journals is non transparent in psychology. Is there any evidence for this claim or just the authors opinion? To my knowledge, there are many societies/universities that have open processes where all candidates that have formal requirements can be considered for the role.
2. The second sentence - Researchers generally accept the outcomes of this opaque decision-making process because it is modus operandi is also a statement that is not grounded on evidence. You should know as psychologists that we cannot know how people feel/think/act unless you asked them.
3. I think that "editors-in-chief are arguably the most powerful people in psychological science." is a too strong conclusion as there are more channels today that were 50 years ago.
4. Point 2 – Editors-in-chief are the face of the field. – is very general and does not refer to psychology specifically.
5. Point 3 - Editors-in-chief choose the associate editors and other board members – what is the evidence of this claim? There are mechanisms in societies/institutions for controlling this. At least in the journals where I am editor. An editor can propose but that does not mean he/she can choose for example 5 members of the board that are male, white and from US (or anywhere else). Many publishing houses have their code of conducts (like Springer <https://www.springernature.com/gp/editors/code-of-conduct-journals>) yet you did not mention any.
6. You repeat that the process of choosing EiC is opaque – in order to claim this you have to have evidence.
7. Recommendations:
 - 2- Good recommendation but also does not guarantee absence of bias and adds more complications in non-profit journals.
 - 3 - Set limits on how many editorial boards a person can serve on at one time. –There are different journals that serve for different purposes, for example in local language (maybe a non-indexed journal) or in English (indexed). Why does a person have to be limited on just one? I understand that people who sit on 10 boards are not doing their job properly but some people are more capable than others (individual differences).
8. "there are widespread systemic barriers in place that prevent women and people of color from advancing up the power hierarchy to the level of full professor". – you need a reference to support the claim. In my country there is no barrier to become a full professor if you are a woman and I live in EU.
9. The conclusion has to be for all readers, not just for those on twitter (X). It is confusing. The whole article is not written just for them.

Reviewer #2 (Remarks to the Author):

This is an opinion piece, based on findings from other authors' surveys. There's not that much one can disagree with here, and even if one did, again, it's an opinion that is stated.

The analogies to selecting a pope or a dodgeball player don't work too well in my opinion. It makes for a somewhat angry and snarky tone. The conclusion also comes off as pretty aggressive IMO. I don't think attacks on people who are working hard within the system to achieve change should be based on their identity, but on their actions. Just my two cents.

Proposal #1 might say more about the new criteria, rather than the old ones we already know well.

Proposal #2 seems to me like not the best idea. Instead, editors can be selected FOR their plans to bring in a diverse team of AEs (who they need to mentor and work with) and a diverse editorial board. As editor of JEP:HPP, I have increased diversity at the racial, gender and geographical level in both groups, because my vision statement said I would and that is part of why the committee chose me.

Proposal #3 My board is selected by my AEs and I and so they reflect our own diversity. Again, board members need to be reviewers that we know and trust, so just picking anyone isn't going to work as well.

RE. lack of eligible candidates, here's a story. When I looked for my last editorial team (this is my 3rd term as editor of an APA journal), instead of just asking people I could think of, I sent a general call for self nomination (including to the SPARK society). For every woman who nominated herself, I received 5 self-nominations from men. By using this process, I was still able to choose as many women as men, and found many individuals I would never have considered.

Re. a pipeline - I started an intern program whereby I train and pay some post docs (selected for diversity) to work with me on the editorial pre-external review screening process. This is a training that is much closer to editorial work and will prepare them better to play a bigger role in the process later on.

In any case, .I just wanted to give examples from my own experience because I think it at least illustrates that there are people in the system who care, and the current system allows for change. There is so much discussion among APA editors for increasing diversity - and there are many programs (mine are examples) that are already creating improvements. To recognize where improvements are being made could also be a (more positive) way to encourage people in this direction.

warmly

Isabel Gauthier, Editor, JEP:HPP

Enclosed please find our revised manuscript entitled, “Diversity at the Level of Editor-in-Chief Remains Elusive” for consideration as a Comment in *Communications Psychology*. In this letter, we describe the changes that were made to address the issues raised in the reviews.

We addressed each of Reviewer 1’s comments as follows:

Reviewer 1: I have read your comment with great interest. The title is directed towards psychology but the whole content of the comment can be discussed in any science. I agree with most of your points but I think some of them have to be supported with more evidence. I suggest major revision, you may find my comments below:

Our Response: We appreciate the positive feedback.

1. Reviewer 1: The first sentence is aluding that the process of choosing editor in chief in journals is non transparent in psychology. Is there any evidence for this claim or just the authors opinion? To my knowledge, there are many societies/univeristies that have open processes where all candidates that have formal requirements can be considered for the role.

Our Response: We now make clear this is our opinion. Also, it is not necessarily the criteria for applying to the position that is opaque, but rather the decision-making process.

2. Reviewer 1: The second sentence - Researchers generally accept the outcomes of this opaque decision-making process because it is modus operandi is also a statement that is not grounded on evidence. You should know as psychologists that we cannot know how people feel/think/act unless you asked them.

Our Response: We deleted this sentence.

3. Reviewer 1: I think that “editors-in-chief are arguably the most powerful people in psychological science.” is a too strong conclusion as there are more channels today that were 50 years ago.

Our Response: We deleted this sentence.

4. Reviewer 1: Point 2 – Editors-in-chief are the face of the field. – is very general and does not refer to psychology specifically.

Our Response: We agree and have revised this sentence to state they are the face of “science.”

5. Reviewer 1: Point 3 - Editors-in-chief choose the associate editors and other board members – what is the evidence of this claim? There are mechanisms in societies/institutions for controlling this. At least in the journals where I am editor. An editor can propose but that does not mean he/she can choose for example 5 members of the board that are male, white and from US (or anywhere else). Many publishing houses have their code of conducts (like Springer <https://www.springernature.com/gp/editors/code-of-conduct-journals>) yet you did not mention any.

Our Response: We agree that there may be constraints on the composition of associate editors, but candidates are still put forth by the editors-in-chief. The constraints may have some influence on very overt bias, but likely cannot mitigate the influence of factors such as ideological bias, familiarity, and other forms of homophily.

6. Reviewer 1: You repeat that the process of choosing EiC is opaque – in order to claim this you have to have evidence.

Our Response: We make clear that this is our opinion. Differences in opinion about the opacity may reflect familiarity with the system - i.e., those that serve or have served as EiC may be one group that has more insight into the process. That said, this small group is not necessarily the primary audience for the commentary. Indeed, we suspect that this is the group that may present the most push back against our recommendations. Our commentary reflects the views/experiences of the much larger group of researchers who are not (and will never be) EiCs.

7. Reviewer 1: Recommendations:
 2- Good recommendation but also does not guarantee absence of bias and adds more complications in non-profit journals.
 3 - Set limits on how many editorial boards a person can serve on at one time. – There are different journals that serve for different purposes, for example in local language (maybe a non-indexed journal) or in English (indexed). Why does a person have to be limited on just one? I understand that people who sit on 10 boards are not doing their job properly but some people are more capable than others (individual differences).

Our Response: The rationale for limiting the number of editorial boards is not due to competence or ability to take-on a large amount of work, but rather it is an issue of equity and to increase the number of researchers participating in the process (i.e., to spread the wealth, so to speak).

8. Reviewer 1: “there are widespread systemic barriers in place that prevent women and people of color from advancing up the power hierarchy to the level of full professor”. – you need a reference to support the claim. In my country there is no barrier to become a full professor if you are a woman and I live in EU.

Our Response: We are glad to hear that there may be a lack of barriers in your country. However, the data would indicate that your situation is not representative of psychology or science more generally. In the revision, we softened the language of this statement and provide a citation.

9. Reviewer 1: The conclusion has to be for all readers, not just for those on twitter (X). It is confusing. The whole article is not written just for them.

Our Response: We did not intend to imply this was for only those on Twitter. We revised this section.

We addressed each of Reviewer 2’s comments as follows:

1. Reviewer 2: This is an opinion piece, based on findings from other authors' surveys. There's not that much one can disagree with here, and even if one did, again, it's an opinion that is stated.

Our Response: We appreciate the thoughtful feedback and have tried to incorporate as many revisions as possible while maintaining a strong viewpoint that will hopefully inspire discussion and further debate. Our approach is that disagreement is useful and crafting an opinion piece to which everyone agrees is not as useful as one that generates discussion.

2. Reviewer 2: The analogies to selecting a pope or a dodgeball player don't work too well in my opinion. It makes for a somewhat angry and snarky tone. The conclusion also comes off as pretty aggressive IMO. I don't think attacks on people who are working hard within the system to achieve change should be based on their identity, but on their actions. Just my two cents.

Our Response: We deleted the dodgeball analogy. We decided to keep the pope analogy because we believe that it is important to engage the reader (even if due to initial disagreement) and grab their interest enough to keep reading the article.

3. Reviewer 2: Proposal #1 might say more about the new criteria, rather than the old ones we already know well.

Our Response: We added additional information about new criteria options (p. 5).

4. Reviewer 2: Proposal #2 seems to me like not the best idea. Instead, editors can be selected FOR their plans to bring in a diverse team of AEs (who they need to

mentor and work with) and a diverse editorial board. As editor of JEP:HPP, I have increased diversity at the racial, gender and geographical level in both groups, because my vision statement said I would and that is part of why the committee chose me.

Our Response: We applaud the work you have done as editor at JEP:HPP. And there are certainly numerous anecdotes and singular cases in which there are EICs working hard to increase diversity. That said, the overall data for the field shows that these cases are not the norm and psychology continues to lack diversity at the top level.

We agree that it is necessary that editors be chosen for their plans for diversity, but it is not sufficient in our opinion. It is important to have diversity at the top level of the hierarchy given the power of this position.

5. Reviewer 2: Proposal #3 My board is selected by my AEs and I and so they reflect our own diversity. Again, board members need to be reviewers that we know and trust, so just picking anyone isn't going to work as well.

Our Response: The issue of how well our proposal will work is an empirical question. We understand that the desire of EICs will be to choose and work with AEs they like and trust. But we contend that science is stronger when homophily is reduced and adversarial collaborations exist. The data on the peer review system would indicate that the current approach has not been effective on several levels, so it is worth pursuing new ideas even if not popular.

6. Reviewer 2: RE. lack of eligible candidates, here's a story. When I looked for my last editorial team (this is my 3rd term as editor of an APA journal), instead of just asking people I could think of, I sent a general call for self nomination (including to the SPARK society). For every woman who nominated herself, I received 5 self-nominations from men. By using this process, I was still able to choose as many women as men, and found many individuals I would never have considered.

Our Response: We wish that all EICs would take your approach to increasing diversity in the peer review process. Unfortunately, data indicates that your approach and success is not the norm.

7. Reviewer 2: Re. a pipeline - I started an intern program whereby I train and pay some post docs (selected for diversity) to work with me on the editorial pre-external review screening process. This is a training that is much closer to editorial work and will prepare them better to play a bigger role in the process later on.

Our Response: While we applaud the idea to create a pipeline of junior investigators from diverse backgrounds for editorial roles, we do have some concerns with the internal training program approach. Although gaining

experience is useful, one of the purposes of increasing diversity is to create change and have unique thoughts and perspectives in the peer review system. Given the lack of diversity at the top level, these training programs may serve to decrease diversity in ideology and approaches to peer review, thus, maintaining the status quo.

8. Reviewer 2: In any case, I just wanted to give examples from my own experience because I think it at least illustrates that there are people in the system who care, and the current system allows for change. There is so much discussion among APA editors for increasing diversity - and there are many programs (mine are examples) that are already creating improvements. To recognize where improvements are being made could also be a (more positive) way to encourage people in this direction.

Our Response: In the revision, we acknowledge that there are many individuals doing excellent work in increasing diversity in the peer review system (p. 4). But the structural system and hierarchy of power continues to be homogenous, particularly at the top level.

We appreciate the thoughtful and constructive feedback we received about the manuscript. We believe the revised manuscript is improved over the original submission, and we hope that it is now suitable for publication. Please let us know if you would like us to make additional revisions, or if you would like us to provide any additional information.

Sincerely,

Gerald J. Haeffel
ghaeffel@nd.edu

Zhicheng Lin
The Chinese University of Hong Kong, Shenzhen

Adeyemi Adetula
Université Grenoble Alpes

Ivan Vargas
University of Arkansas

Jenalee R. Doom
University of Denver

Yusuke Moriguchi
Kyoto University

Ana J. Bridges

University of Arkansas

Willie R. Cobb
New York City, NY, United States

29th Nov 23

Dear Jerry,

Thank you for your patience during the editorial evaluation process.

I have reviewed the revisions you undertook to your manuscript "Diversity at the Level of Editor-in-Chief Remains Elusive in Psychological Science" in response to the previous reviewer concerns.

We remain very interested in the possibility of publishing your Comment in Communications Psychology, but would like to consider your response to a list of concerns in the form of a revised manuscript before we make a decision on publication. In particular, I stress that some arguments in the present version of the manuscript need to be substantiated with - or reconsidered in light of - further evidence.

To aid you with that task, I have included a marked-up version of your manuscript.

In sum, we invite you to revise your manuscript taking into account all editor comments.

EDITORIAL POLICIES AND FORMATTING

You will find a complete list of formatting requirements following this link:

<https://www.nature.com/documents/commsj-style-formatting-checklist-comment.pdf>

Please use the checklist to prepare your manuscript for resubmission.

If you have any questions about any of our policies or formatting, please don't hesitate to contact me.

Please use the following link to submit your revised manuscript and a point-by-point response to the referees' comments (which should be in a separate document to any cover letter):

[link redacted]

We hope to receive your revised paper within 4 weeks; please let us know if you aren't able to submit it within this time so that we can discuss how best to proceed. If we don't hear from you, and the revision process takes significantly longer, we may close your file.

Please do not hesitate to contact me if you have any questions or would like to discuss these revisions further. We look forward to seeing the revised manuscript and thank you for the opportunity to review your work.

Best wishes,

Marike

Marike Schiffer, PhD
Chief Editor
Communications Psychology

5th Jan 24

Dear Jerry,

I have read your revised Comment titled "Psychology Needs More Diversity at the Level of Editor-in-Chief" and I am delighted to say that we are happy, in principle, to publish it in Communications Psychology under a Creative Commons 'CC BY' open access license.

We will not send your revised paper for further review. If the revised paper is in Communications Psychology format, in an accessible style, and of appropriate length, we shall accept it for publication immediately. I have attached an edited version of your manuscript, and ask you to attend to each comment for a final round of edits.

EDITORIAL REQUESTS:

* Please review the changes in the attached copy of your manuscript, which has been edited for style, and address the comments and queries I have added. If using Word, please use the 'track changes' feature to make the process of accepting your manuscript more efficient.

*Please ensure that you address all requests in the attached Editorial Requests Table.

* Communications Psychology uses a transparent peer review system. On author request, confidential information and data can be removed from the published reviewer reports and rebuttal letters prior to publication. If you are concerned about the release of confidential data, please let us know specifically what information you would like to have removed. Please note that we cannot incorporate redactions for any other reasons.

*If you have not done so already, please alert me to any related manuscripts from your group that are under consideration or in press at other journals, or are being written up for submission to other journals (see www.nature.com/authors/editorial_policies/duplicate.html for details).

FORMATTING GUIDELINES:

Please use the checklist to prepare your manuscript for final submission. In the following, I also highlight some issues of particular importance.

* References

References appear as superscript Arabic numerals, in order of mention. The reference list mentions references in the numerical order in which they are mentioned in the main text. If a reference is cited more than once, the same number is used throughout the text and the reference receives a single entry in the reference list.

Only papers that have been published or accepted by a named publication should be in the reference list (preprints and citations of datasets are also permitted). Unpublished/Submitted research should not be included in the reference list; it should only be mentioned briefly and parenthetically in the main text. Note that no major arguments should rely on unpublished research.

Published conference abstracts and URLs for websites should be cited parenthetically in the text, not in the reference list.

* Competing interests

Please include a "Competing interests" statement after the References. Note that we ask authors to declare both financial and non-financial competing interests. For more details, see <https://www.nature.com/authors/policies/competing.html>. If you have no financial or non-financial competing interests, please state so: "The authors declare no competing interests."

In order to accept your paper, we require the following:

* A cover letter describing your response to our editorial requests.

* The final version of your text as a Word or TeX/LaTeX file, with any tables prepared using the Table menu in Word or the table environment in TeX/LaTeX and using the 'track changes' feature in Word.

At acceptance, the corresponding author will be required to complete an Open Access Licence to Publish on behalf of all authors, declare that all required third-party permissions have been obtained.

Please note that your paper cannot be sent for typesetting to our production team until we have received this information; **therefore, please ensure that you have this ready when submitting the final version of your manuscript.**

ORCID

Communications Psychology is committed to improving transparency in authorship. As part of our efforts in this direction, we are now requesting that all authors identified as 'corresponding author' create and link their Open Researcher and Contributor Identifier (ORCID) with their account on the Manuscript Tracking System (MTS) prior to acceptance. ORCID helps the scientific community achieve unambiguous attribution of all scholarly contributions. For more information please visit <http://www.springernature.com/orcid>

For all corresponding authors listed on the manuscript, please follow the instructions in the link below to link your ORCID to your account on our MTS before submitting the final version of the manuscript. If you do not yet have an ORCID you will be able to create one in minutes.

IMPORTANT: All authors identified as 'corresponding author' on the manuscript must follow these instructions. Non-corresponding authors do not have to link their ORCIDs but are encouraged to do so. Please note that it will not be possible to add/modify ORCIDs at proof. Thus, if they wish to have their ORCID added to the paper they must also follow the above procedure prior to acceptance.

To support ORCID's aims, we only allow a single ORCID identifier to be attached to one account. If you have any issues attaching an ORCID identifier to your MTS account, please contact the Platform Support Helpdesk.

[link redacted]

We hope to hear from you within two weeks; please let us know if the process may take longer.

Best wishes,

Marike
Marike Schiffer, PhD
Chief Editor
Communications Psychology